# Automata-based constraints for language model decoding

**Terry Koo**∗**, Frederick Liu**∗**, and Luheng He**
Google DeepMind
{terrykoo,frederickliu,luheng}@google.com

## Abstract

Language models (LMs) are often expected to generate strings in some formal language; for example, structured data, API calls, or code snippets. Although LMs can be tuned to improve their adherence to formal syntax, this does not *guarantee* conformance, especially with smaller LMs suitable for large-scale deployment. In addition, tuning requires significant resources, making it impractical for uncommon or task-specific formats. To prevent downstream parsing errors we would ideally *constrain* the LM to only produce valid output, but this is severely complicated by tokenization, which is typically both ambiguous and misaligned with the formal grammar. We solve these issues through the application of automata theory, deriving an efficient closed-form solution for the *regular languages*, a broad class of formal languages with many practical applications, including API calls or schema-guided JSON and YAML. We also discuss pragmatic extensions for coping with the issue of high branching factor, and extend our techniques to *deterministic context-free languages*, which similarly admit an efficient closed-form solution. Previous work on this topic (Willard & Louf, 2023) layers bespoke solutions onto automata, leading to problems with speed, correctness, and extensibility. Instead, we reformulate the entire task in terms of automata so we can leverage well-studied and well-optimized algorithms. Our system compiles constraints ~7,000x faster, is provably correct, and can be extended in a modular fashion.

## 1 Introduction

A common use case for LMs is generating output in some formal language (Liu et al., 2024); for example, structured data, API calls (Yao et al., 2023), or code snippets (Li et al., 2022). While powerful LMs often produce syntactically well-formed output, this is not guaranteed. This paper describes methods for *constraining* an LM to generate broad classes of formal languages. One general recipe for applying constraints to any LM is to mask the decoder logits (Deutsch et al., 2019; Zhang et al., 2019):

1. Based on the current state of the constraint, build a mask of valid next tokens.
2. Penalize the sampling logits using the mask, so only valid tokens are considered.
3. Feed the selected token back to the constraint, to update its state for the next step.

Tokenization is a major problem because popular LMs use data-driven sub-word tokenizers (Sennrich et al., 2016; Kudo & Richardson, 2018) whose segmentations are generally both ambiguous and misaligned with formal-language tokens. For example, consider an API call like `foo(x="bar")`, which is typically lexed as `foo|(|x|=|"bar"|)`. An LM might tokenize this as `foo(|x="|ba|r|")`, merging some lexer tokens and splitting others; see Appendix A.

Naively, we could force the tokenization to align with the formal syntax, but this can harm quality (Lundberg, 2023). On the other hand, accepting a token like `x="` involves recognizing a variable name, operator, and the start of a string literal in one step. Mapping a formal language constraint onto an arbitrary LM tokenizer involves a long tail of such special cases.

We find elegant solutions to these issues in automata theory (Hopcroft et al., 2001; Riley et al., 2009). Our main contributions are primarily conceptual rather than empirical:

1. **Identify** an as-yet unnoticed connection between detokenization and transduction.
2. **Solve** the tokenization issues using this connection and operations on automata.
3. **Define** extensions that address practical problems of efficiency and convenience.

---

∗Equal contribution, alphabetical.

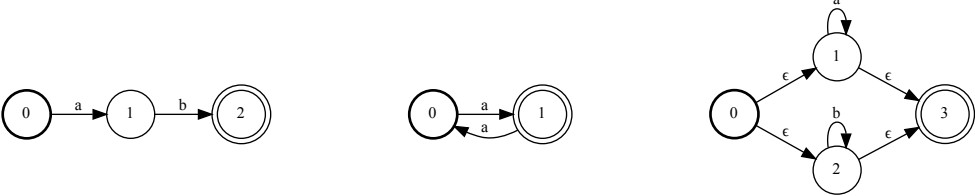

Figure 1: FSAs that accept ab (left), odd numbers of as (center), and runs of as or bs (right). States are depicted as circles, with the start state in bold and final states doubled. Edges are depicted as directed arcs, labeled with the relevant input symbol, or $\epsilon$ if none.

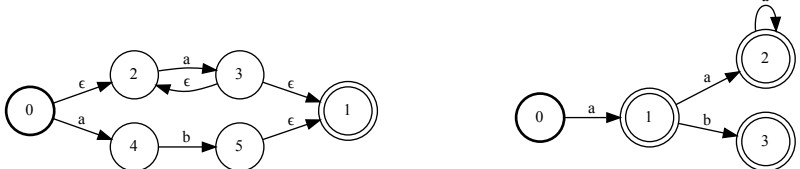

Figure 2: The FSA constructed from the regular expression /a+|ab/ is initially non-deterministic (left), but can be determinized (right).

Prior work (Willard & Louf, 2023) has addressed these tokenization issues, but with bespoke solutions. By developing a proper theoretical grounding, we gain advantages in speed (see Section 6.1), correctness (see Appendix B), and extensibility (see Section 3.2 and 4.3).

## 2 Finite-state constraints

In this section, we provide some background and then describe our main set of contributions.

### 2.1 Finite-state automata (FSAs)

A *finite-state automaton* $\mathcal{A}$ is a tuple $(\Sigma, Q, I, F, E)$ where $\Sigma$ is a set of input symbols, $Q$ is a finite set of states, $I \in Q$ and $F \subseteq Q$ are initial and final states, and $E \subseteq Q \times \Sigma_\epsilon \times Q$, where $\Sigma_\epsilon = \Sigma \cup \{\epsilon\}$, is a set of edges (Hopcroft et al., 2001; Riley et al., 2009). Each edge $e \in E$ is a tuple $(e^s, e^\sigma, e^t)$ of source state $e^s$, input label $e^\sigma$ or $\epsilon$ if none, and target state $e^t$. See Figure 1.

An FSA $\mathcal{A}$ *accepts* $w \in \Sigma^*$ if there exist $e_1, ..., e_n \in E$ such that $w = e_1^\sigma \cdots e_n^\sigma$, $e_1^s = I$, $e_n^t \in F$, and $e_i^s = e_{i-1}^t$ for $i > 1$; note that $n > |w|$ iff $\exists i, e_i^\sigma = \epsilon$. To express the functional behavior of FSAs, we overload notation and define $\mathcal{A}(w)$ as a predicate that is true iff $\mathcal{A}$ accepts $w$. For any FSA $\mathcal{A}$, we define its *language* $L_\mathcal{A} = \{w \in \Sigma^* : \mathcal{A}(w)\}$ as the set of strings it accepts. More generally, the *regular languages* can be defined as $\{L_\mathcal{A} : \mathcal{A}$ is an FSA$\}$ (Chomsky, 1956).

Conveniently, the regular languages can also be defined by regular expressions[1], which are equivalent to FSAs (Kleene, 1951). Common tools like UNIX grep compile regular expressions into FSAs with $\Sigma =$ Unicode, which are then used as predicates on text (McNaughton & Yamada, 1960; Thompson, 1968). See Figure 2 (left).

An FSA is *deterministic* if $\forall e \in E$, $e^\sigma \neq \epsilon$ and $\forall q \in Q, a \in \Sigma$, $|\{e \in E : e^s = q \land e^\sigma = a\}| \leq 1$. Intuitively, outbound edges have unique inputs so executing the FSA is trivial: track the current state and traverse the edge matching the next input. Surprisingly, non-deterministic and deterministic FSAs are equivalent. In particular, given an arbitrary FSA $\mathcal{A}$ one can build a deterministic FSA $\mathcal{A}'$ such that $L_\mathcal{A} = L_{\mathcal{A}'}$ (Rabin & Scott, 1959). See Figure 2.

### 2.2 Finite-state transducers (FSTs)

A *finite-state transducer* is an FSA that generates output. Formally, an FST $\mathcal{T}$ is a tuple $(\Sigma, \Delta, Q, I, F, E)$ where $\Sigma, Q, I$, and $F$ are as defined for FSAs, $\Delta$ is a set of output symbols, and $E \subseteq Q \times \Sigma_\epsilon \times \Delta_\epsilon \times Q$ is a set of edges (Mohri, 1997; 2004; Riley et al., 2009). Each edge $e \in E$ is a tuple $(e^s, e^\sigma, e^\delta, e^t)$ where $e^s, e^\sigma$, and $e^t$ are as defined for FSAs and $e^\delta$ is its output label, or $\epsilon$ if none. See Figure 3.

---

[1]In this paper we use the mathematical definition of regular expressions. Many tools extend regular expressions with non-regular features like backreferences—at the cost of exponential runtime.

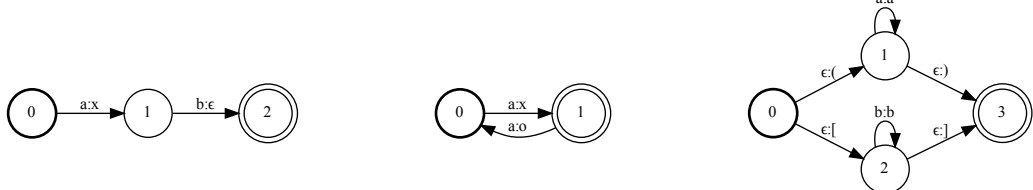

Figure 3: FSTs that transduce ab into x (left), odd numbers of as into xoxo ⋯ x (center), and runs of as or bs into bracketed versions of themselves (right). Edge labels are $e^\sigma{:}e^\delta$.

| **Tokens** |
| --- |
| f |
| oo |
| foo |
| for |
| food |

Figure 4: A simple vocabulary of tokens (left), and a detokenizing FST that transduces sequences of those tokens into sequences of characters (right).

An FST $\mathcal{T}$ *transduces* $w \in \Sigma^*$ into $v \in \Delta^*$ if there exist $e_1, ..., e_n \in E$ such that $w = e_1^\sigma \cdots e_n^\sigma$, $v = e_1^\delta \cdots e_n^\delta$, $e_1^s = I$, $e_n^t \in F$, and $e_i^s = e_{i-1}^t$ for $i > 1$. Similar to FSAs, we write[2] $v = \mathcal{T}(w)$ if $\mathcal{T}$ transduces $w$ to $v$. Critically, the output of one FST can be fed as the input of another FST or FSA (Riley et al., 2009). Given FSTs $\mathcal{T}_1$ and $\mathcal{T}_2$ where $\Delta_1 = \Sigma_2$, we can *compose* them into a new FST $\mathcal{T}' = \mathcal{T}_2 \circ \mathcal{T}_1$ where $\Sigma' = \Sigma_1$, $\Delta' = \Delta_2$, and $\mathcal{T}'(w) = \mathcal{T}_2(\mathcal{T}_1(w))$. Similarly, given an FST $\mathcal{T}_1$ and FSA $\mathcal{A}_2$ where $\Delta_1 = \Sigma_2$, we can compose them into a new FSA $\mathcal{A}' = \mathcal{A}_2 \circ \mathcal{T}_1$ where $\Sigma' = \Sigma_1$ and $\mathcal{A}'(w) = \mathcal{A}_2(\mathcal{T}_1(w))$.

### 2.3 Detokenization as transduction

Our first contribution is a reformulation of detokenization (i.e., the process of converting token sequences back into text) as an FST, using the following construction:

---
**Algorithm 1** Builds detokenizing FST $\mathcal{T}_V = (\Sigma_V, \Delta_V, Q_V, I_V, F_V, E_V)$ from vocabulary $V$

---
$\Sigma_V \leftarrow V$, $\Delta_V \leftarrow \{v_i : v \in V, 1 \le i \le |v|\}$ ▷ inputs/outputs are tokens/characters of $V$
$Q_V \leftarrow \{q_r\}$, $I_V \leftarrow q_r$, $F_V \leftarrow \{q_r\}$, $E_V \leftarrow \{\}$ ▷ initially, $\mathcal{T}_V$ only has a root state $q_r$
**for** $v \in V$ **do**
    $q_0 \leftarrow q_r$, $n \leftarrow |v|$
    **for** $i = 1$ to $n - 1$ **do** ▷ build a chain for all but the last character
        $Q_V \leftarrow Q_V \cup \{q_i\}$, $E_V \leftarrow E_V \cup \{(q_{i-1}, \epsilon, v_i, q_i)\}$ ▷ no input with non-final output
    $E_V \leftarrow E_V \cup \{(q_{n-1}, v, v_n, q_r)\}$ ▷ input whole token with last output, cycle to root

---

For compactness, common prefixes of the chains can be merged to form a trie-like structure, as in Figure 4; see Appendix B.1 for a proof of correctness.

### 2.4 Adapting regular expressions to tokens

Our next contribution is a generic method for adapting any FSA from characters to tokens. Specifically, given a token vocabulary $V$ and an FSA $\mathcal{A}$ that accepts character sequences, $\mathcal{A}' = \mathcal{A} \circ \mathcal{T}_V$ accepts essentially the same language as $\mathcal{A}$, but in token form. More precisely, for each token sequence $w \in L_{\mathcal{A}'}$, the detokenization of $w$ is in $L_{\mathcal{A}}$.

Note that the converse does not hold: $w \in L_{\mathcal{A}}$ has a counterpart in $L_{\mathcal{A}'}$ only if $w$ can be segmented into tokens from $V$. For example, suppose $\mathcal{A}$ accepts any number in hexadecimal format, but the tokens in $V$ only cover the digits 0-9. Nevertheless, to the extent that $L_{\mathcal{A}}$ can be tokenized by $V$, strings in $L_{\mathcal{A}}$ are represented in $L_{\mathcal{A}'}$. Indeed, due to tokenization

---
[2]$\mathcal{T}(w)$ should be a set because a non-deterministic FST can transduce the same input to different outputs. In this paper, the FSTs we define satisfy $|\mathcal{T}(w)| = 1$ so we simply write $v = \mathcal{T}(w)$.

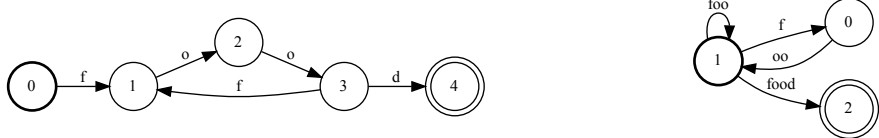

Figure 5: The character-based FSA equivalent to `/(foo)+d/` (left) and its composition with the detokenizing FST from Figure 4 (right). Note that the same text can have many tokenizations (e.g., `foo` vs `f`|`oo`), and tokens are allowed to cross sub-expression boundaries (e.g., `food` merges the last repeat of `/(foo)+/` with `/d/`).

ambiguity, there may be multiple token sequences in $L_{\mathcal{A}'}$ that are equivalent to the same string in $L_{\mathcal{A}}$. See Figure 5.

We now present our method for constraining an LM to a regular language:

---

**Algorithm 2** Constrains LM $L$ with vocabulary $V$ to generate the language of regex $R$

$\mathcal{T}_V \leftarrow \text{BuildDetokenizingFST}(V)$      ▷ token-to-character FST, see Algorithm 1
$\mathcal{A}_R \leftarrow \text{BuildRegexFSA}(R)$      ▷ character-accepting FSA (Thompson, 1968)
$\mathcal{A}_{R \circ V} \leftarrow \text{Determinize}(\mathcal{A}_R \circ \mathcal{T}_V)$      ▷ token-accepting FSA
$q \leftarrow I_{R \circ V}$      ▷ start from initial FSA state
**for** $t = 1$ to $T$ **do**      ▷ decoding steps
    $\ell \leftarrow \text{ComputeLogits}(L)$
    $A \leftarrow \{e^{\sigma} : e \in E_{R \circ V} \wedge e^s = q\}$      ▷ allowed next tokens
    **for** $i = 1$ to $|V|$ **do**      ▷ penalize logits as in Deutsch et al. (2019)
        **if** $v_i \notin A$ **then** $\ell_i \leftarrow -\infty$
    $\hat{v} \leftarrow \text{SampleNextToken}(L, \ell)$
    $\hat{e} \leftarrow e$ s.t. $e \in E_{R \circ V} \wedge e^s = q \wedge e^{\sigma} = \hat{v}$      ▷ find the matching edge
    $q \leftarrow \hat{e}^t$      ▷ traverse the edge

---

Note that $\mathcal{A}_{R \circ V}$ is a closed-form solution: it expresses $R$ using all relevant tokens from $V$ and can be executed independently from both. See Appendix B.2 for a proof of correctness.

The required operations are indexing, slicing, and basic arithmetic, which are efficient and simple enough to execute directly on an accelerator with minimal latency overhead. One pleasing aspect of this solution is how neatly it separates concerns amongst the two halves:

- $\mathcal{T}_V$ is vocabulary-specific and, while large, can easily be pre-computed for each LM.
- $\mathcal{A}_R$ is vocabulary-agnostic, easily specified, and portable across different LMs.

This clean decomposition is only possible because FST-FSA composition provides a fast, automatic, and general method for joining the two halves.

For example, alternative detokenization automata (see Section 4.3) can be slotted into $\mathcal{T}_V$ without changing the rest of the system. Similarly, alternative constraint automata (see Section 3.1) can be substituted for $\mathcal{A}_R$ and FST composition still works.

## 2.5 Extensions

Our last contribution in this section is a set of regular expression *extensions*, written as specially-named capturing groups, that greatly increase the efficiency and expressiveness of the system. We describe some illustrative examples below and list others in Table 1.

### 2.5.1 Wildcard matching

One problem in our approach occurs with "wildcard" matches like `/./` or `/[^0-9]/` that match nearly any character. After composition with $\mathcal{T}_V$, some states in the resulting FSA will have almost $|V|$ outbound edges, making it expensive to use—keep in mind that $|V| > 100k$ for commonly-used LMs (Willard & Louf, 2023, Section 3 describes a similar issue).

We mitigate this issue by defining *terminal labels*, which are token IDs disjoint from $V$ that map to pre-computed masks of valid tokens. For example, `/(?P<PARAGRAPH_TOKEN>)/` parses into a terminal edge whose mask indicates newline-free tokens. This is useful for structuring

| Name | Description |
|------|-------------|
| `QUOTED_TEXT` | Matches a quoted string with backslash escapes. |
| `UNQUOTED_TEXT` | Matches a YAML-style non-quoted string. |
| `IMAGE` | Matches an image generated by a multi-modal LM. |
| `TEXT_TOKEN` | Matches a single text token. |
| `PARAGRAPH_TOKEN` | Matches a single text token with no newlines. |
| `TEXT_UNTIL` | Matches text tokens until a stop phrase appears. |
| `SUBSTRING_OF` | Matches any substring of a given string. |
| `DELIMITED_LIST` | Matches a delimited list of homogeneous items. |
| `DELIMITED_SUBSEQUENCE_OF` | Matches a delimited subset of a heterogeneous list. |

Table 1: A sampling of available extensions. Those above the line are wildcard matchers (see Section 2.5.1), while those below are syntactic sugar (see Section 2.5.2)
.

free text: for example, `/Summary:(\n\* (?P<PARAGRAPH_TOKEN>)+){3,5}/` would match the heading "Summary:" followed by three to five bullets.

When penalizing logits, terminal masks are applied en masse. When updating state, if the selected token matches a terminal mask we traverse that terminal edge.

Note that a normal token edge and terminal edge can both match the sampled token, leading to ambiguity about which to traverse. Similarly, two terminal edges can match the sampled token. We address this with a simple heuristic: prefer normal token edges when available, and otherwise follow a semi-arbitrary precedence ordering over terminal labels. This heuristic has worked well because terminals are generally used for wildcard matches, and in practice wildcards are typically abutted by disjoint delimiting expressions. For example, in the bulleted list constraint above, the `/(?P<PARAGRAPH_TOKEN>)+/` expression is bounded by a newline.

### 2.5.2 Syntactic sugar

Sometimes, a constraint is inefficient to define as a regular expression. For example, suppose we wish to reduce hallucination by forcing the LM to generate a substring of some reference $u$, such as a user utterance. Unfortunately, the regular expression for substrings of $u$ is $O(|u|^2)$, even though the corresponding FSA is $O(|u|)$—this can be viewed as a defect of the specification language.

We therefore define the syntactic sugar `/(?P<SUBSTRING_OF>abc)/` to match any substring of "abc", resolving the defect by providing a $O(|u|)$ specification syntax. Besides improving usability, these extensions mitigate the explosion in regular expression size that can occur in complex applications like JSON constraints. In some cases, growth in the regular expression reflects growth in the resulting FSA, but we have found that they diverge in some important practical use cases.

## 3 Push-down constraints

This section briefly presents our second set of contributions, describing how our system is easily extended to grammar-based constraints using push-down automata. We begin with some background on PDAs and then describe our contributions.

### 3.1 Push-down automata (PDAs)

A *push-down automaton* can be viewed as an FSA equipped with a stack. Formally, a PDA $\mathcal{P}$ is a tuple $(\Sigma, \Pi, S, Q, I, F, E)$ where $\Sigma$, $Q$, $I$, and $F$ are as defined for FSAs, $\Pi$ is a set of stack symbols, $S \in \Pi$ is a unique initial stack symbol, and $E \subseteq Q \times \Sigma_\epsilon \times \Pi^* \times Q \times \Pi^*$ is a set of edges (Hopcroft et al., 2001). Each edge $e \in E$ is a tuple $(e^s, e^\sigma, e^\uparrow, e^t, e^\downarrow)$ where $e^s, e^\sigma$, and $e^t$ are as defined for FSAs, and $e^\uparrow$ and $e^\downarrow$ are popped and pushed stack symbols. See Figure 6.

$\mathcal{P}$ accepts $w \in \Sigma^*$ if there exist $e_1, ..., e_n \in E$ such that $w = e_1^\sigma \cdots e_n^\sigma$, $e_1^s = I$, $e_n^t \in F$, $e_i^s = e_{i-1}^t$ for $i > 1$, and $e_i^\uparrow$ and $e_i^\downarrow$ form a coherent sequence of stack edits. Formally, let $s^i$ be the stack before $e_i$ where $s^1 = S$, $n_i = |s^i|$, $m_i = n_i - |e_i^\uparrow|$, and $s^{i+1} = s_{1:m_i}^i e_i^\downarrow$, then $e_i^\uparrow = s_{m_i+1:n_i}^i$ must

| Rules |
| --- |
| S -> /ab/ |
| S -> /a/ S /b/ |

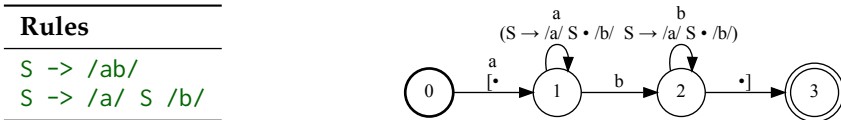

Figure 6: A grammar for the canonical context-free language $a^n b^n$ (left), and an equivalent PDA (right). Edge labels have $e^\sigma$ on top, then $e^\uparrow$ marked with ) and $e^\downarrow$ marked with (. The stack symbols are Earley-style dotted rules (Earley, 1970) denoting "return addresses" (Allauzen & Riley, 2012, Section 4.5). The initial stack symbol $S$ is written as a dot marked with square brackets [ and ].

| Tokens |
| --- |
| a |
| b |
| bb |
| aaab |

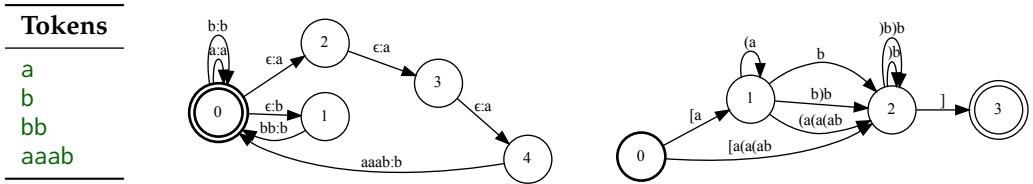

Figure 7: A simple vocabulary of tokens (left), the detokenizing FST built from it (center), and its composition with the PDA from Figure 6 (right). Note that edges are allowed to cross the boundaries of terminals and non-terminals, allowing them to push or pop multiple stack symbols. For legibility, the stack symbols have been simplified into single brackets that are interleaved with the characters of the token.

hold. In other words, at each step $e_i^\uparrow$ must match the suffix of $s_i$, and we update the stack by popping $e_i^\uparrow$ off and pushing $e_i^\downarrow$ on. As above, we write $\mathcal{P}(w)$ to indicate that $\mathcal{P}$ accepts $w$.

Unlike FSAs, non-deterministic and deterministic PDAs are not equivalent: the former accept the *context-free languages*, which are also defined by context-free grammars (Chomsky, 1956; Hopcroft et al., 2001), while the latter accept a strict subset, the *deterministic context-free languages* (Ginsburg & Greibach, 1966). This subset nevertheless includes all regular languages and the LL($k$) and LR($k$) languages (Knuth, 1965), covering the syntax of most programming languages. As a result of this non-equivalence, there is no general algorithm to determinize an arbitrary PDA, as we had for FSAs. Instead, as LL($k$) and LR($k$) parsers do, our system detects non-deterministic grammars and signals an error, leaving it to the author of the grammar to rewrite the grammar in a deterministic manner.

### 3.2 Adapting grammars to tokens

Although PDAs are more expressive than FSAs, they behave similarly in many ways and, crucially, FSTs and PDAs are composable (Allauzen & Riley, 2012). Formally, given an FST $\mathcal{T}_1$ and PDA $\mathcal{P}_2$ where $\Delta_1 = \Sigma_2$, we can compose them into a new PDA $\mathcal{P}' = \mathcal{P}_2 \circ \mathcal{T}_1$ where $\Sigma' = \Sigma_1$ and $\mathcal{P}'(w) = \mathcal{P}_2(\mathcal{T}_1(w))$. As with FSAs, composition with the detokenizing FST $\mathcal{T}_V$ adapts a character-based PDA into one that accepts tokens in $V$. See Figure 7.

This allows us to reuse Algorithm 2 with minimal modification (see Appendix B.3 for proof):

---

**Algorithm 3** Constrains LM $L$ with vocabulary $V$ to generate the language of grammar $G$

---

$\mathcal{T}_V \leftarrow \text{BUILDDETOKENIZINGFST}(V)$      ▷ token-to-character FST, see Algorithm 1
$\mathcal{P}_G \leftarrow \text{BUILDGRAMMARPDA}(G)$      ▷ character-accepting PDA (Allauzen & Riley, 2012)
$\mathcal{P}_{G \circ V} \leftarrow \mathcal{P}_G \circ \mathcal{T}_V$      ▷ token-accepting PDA
$q \leftarrow I_{G \circ V}, \; s \leftarrow \epsilon$      ▷ start from initial PDA state and empty stack
**for** $t = 1$ to $T$ **do**      ▷ decoding steps
    $\ell \leftarrow \text{COMPUTELOGITS}(L)$
    $A \leftarrow \{e^\sigma : e \in E_{G \circ V} \wedge e^s = q \wedge s \text{ ends with } e^\uparrow\}$      ▷ allowed next tokens
    **for** $i = 1$ to $|V|$ **do**      ▷ penalize logits as in Deutsch et al. (2019)
        **if** $v_i \notin A$ **then** $\ell_i \leftarrow -\infty$
    $\hat{v} \leftarrow \text{SAMPLENEXTTOKEN}(L, \ell)$
    $\hat{e} \leftarrow e \text{ s.t. } e \in E_{G \circ V} \wedge e^s = q \wedge s \text{ ends with } e^\uparrow \wedge e^\sigma = \hat{v}$      ▷ find the matching edge
    $q \leftarrow \hat{e}^t, \; \hat{m} \leftarrow |s| - |\hat{e}^\uparrow|, \; s \leftarrow s_{1:\hat{m}} \hat{e}^\downarrow$      ▷ traverse the edge

---

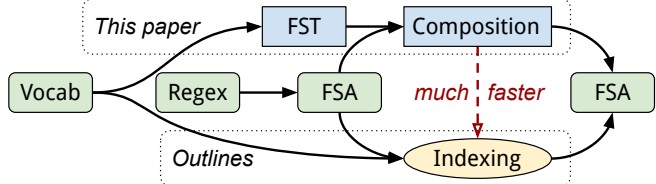

Figure 8: Flowchart comparing this paper with Outlines (Willard & Louf, 2023). Blue boxes are unique to this paper, yellow ovals are unique to Outlines, and green rounded boxes are common to both. Note that composition is many times faster than indexing (see Section 6.1).

It's worth re-emphasizing the benefits of formulating detokenization as an FST. By representing the entire task in terms of automata, our approach easily generalizes from regular expressions to grammars while preserving many desirable features. For example, just as FST-FSA composition enables tokens to cross sub-expression boundaries in the regular expression (see Figure 5), FST-PDA composition enables tokens to cross (non-)terminal boundaries in the grammar (see Figure 7).

## 4 Related work

There is a great deal of work related to the use of automata to constrain learned models.

### 4.1 Automata for sequence models

Automata have been used to constrain learned sequence models for some time. Below, we mention a few representative examples from the literature.

Sproat & Jaitly (2017) and Zhang et al. (2019), among others in this line of work, examine text normalization for TTS applications. Certain phrases are difficult to verbalize (e.g., "4/5" to "four fifths" or "April fifth" or "four out of five [stars]"), and neural models are prone to hallucinations (e.g., "60" to "six"). The authors build FSTs that translate from written forms to possible verbalizations, and use those to guide the deep model similarly to this work. Our approach allows a wider range of constraints, supporting any regular or deterministic context-free language, and also addresses the token-mismatch issues endemic to current LM tokenizers (see Appendix A).

Deutsch et al. (2019) use (intersections of) FSAs and PDAs to constrain the output of a sequential model to some formal language. This work predates the rise of general-purpose LMs, and thus does not address the problems created by mismatches between the LM's tokenization and the targeted formal language. Both Outlines (Willard & Louf, 2023) and this work address those mismatches, but by different means.

### 4.2 Automata for general-purpose LMs

The most relevant work is Outlines (Willard & Louf, 2023), which is based on a bespoke "indexing" operation that builds a token-based overlay on an FSA or PDA. Unfortunately, when applied to PDAs their algorithm is over-constrained w.r.t. tokenization[3] (see Appendix A.1), whereas our approach naturally generalizes to PDAs while preserving tokenization freedom (see Section 3.2). Our approach also easily generalizes to different tokenization schemes (see Section 4.3), which does not seem possible for Outlines and likely requires a rewrite of their indexing algorithm. Finally, their approach is ~7,000x slower to compile than ours (see Section 6.1), so it is only practical when the constraint can be pre-compiled. Our core innovation allows us to recast the entire process in automata-theoretic terms, making our solution extensible (via decomposition into FST and FSA/PDA modules), correct (via basic properties of FSTs), and fast (via access to highly-optimized algorithms). For clarity, Figure 8 illustrates which parts of our system are similar to or different from Outlines.

Building on Outlines, Yin et al. (2024) extended the package with the ability to "compress" runs of text into prefills. This technique is of significant practical interest and could be adapted to our approach as our techniques appear to be largely orthogonal.

---

[3]https://github.com/outlines-dev/outlines/issues/684

SynCode (Ugare et al., 2024) is a more recent system that also exploits FSAs, but handles grammars using LALR(1) and LR(1) parsers rather than PDAs. Like Outlines, their approach relies on a bespoke algorithm; in this case, they speculatively unroll future lexer tokens to allow LM tokens to cross multiple lexer tokens. This introduces significant complexity relative to our purely automata-theoretic approach, and deciding how many lexer tokens to unroll is a trade-off between computational cost and completeness.

Some less closely-related approaches still constrain by masking logits, but dynamically match the vocabulary on each step instead of devising a method to statically pre-compute these matches, as we and the systems above do. For example, Synchromesh (Poesia et al., 2022) iterates the entire LM vocabulary on every decoding step, making it impractical with current LMs. Guidance (Microsoft, 2023) improves upon this with a cached trie, and grammar-constrained decoding (Geng et al., 2023) adds expressive power beyond context-free. While these latter two are more flexible than our approach, they are less efficient and harder to deploy at scale.

Finally, some other approaches no longer predictively mask logits, but instead sample unconstrained continuations and reject invalid ones post-hoc. For example, PICARD (Scholak et al., 2021) converts the top-$k$ tokens to text and performs various levels of validation. Lew et al. (2023) do sequential Monte-Carlo sampling that, in the case of hard constraints, reduces to something like rejection sampling with rescoring. These approaches are quite flexible, but may have issues when the constraints require the LM to select a low-probability token. In addition, note that post-hoc filtering is orthogonal to predictive masking, so our approaches could be merged.

### 4.3 Automata for LM tokenization

Concurrently with review, Berglund et al. (2024) presented a finite-state machine that only accepts "correct" BPE tokenizations. Unlike the simple detokenizing FST presented in Section 2.3, their FSA is *unambiguous*: for any character sequence $w$, it only accepts the correct BPE tokenization of $w$. As they mention in their conclusion, their construction can be straightforwardly generalized into an FST.

From there, we could compose their FST with any regex-derived FSA or grammar-derived PDA, yielding an automaton that accepts only correct BPE tokenizations that also obey the constraint. In essence, any tokenization policy that can be expressed in finite-state form can be dropped into our approach, demonstrating its modularity and generality.

## 5 Applications

The clean design and efficiency of our approach enable a number of different applications. We present several illustrative examples here.

### 5.1 JSON expressions

JSON (Pezoa et al., 2016) is a widely-used structured data format that LMs are often expected to generate. We study the problem of generating JSON that conforms to a *schema*, which is a mapping from field name to type. Field types range from simple primitives like booleans, numbers, and strings, to sub-objects based on sub-schemas, and arrays or unions of other types. Field values can be *nullable*, meaning that they can be substituted with `null`, and fields can be marked *optional*, meaning that both the field name and value may be omitted.

Although the set of all JSON expressions of any kind is a context-free language, somewhat surprisingly the set of JSON expressions conforming to a particular schema is a regular language, as the schema limits recursion. It is difficult to manually write the regular expression corresponding to a particular schema: while leaf-level matchers like `/(true|false)/` are simple enough, complexity grows quickly as they are composed into objects, arrays, and unions. For user convenience, we have implemented tools that automatically translate schemas into regular expressions.

We have also explored schema-free JSON, which as mentioned above is context-free. A simple approach is to impose constant limits $k_O$ and $k_A$ on object and array nesting, which reduces from context-free to regular, and we have implemented tools to generate regular

| Constraint | Compilation speedup | Per-step speedup |
|---|---|---|
| Multiple choice | 7,970x | 29.5x |
| ISO date-time | 7,110x | 24.3x |
| IP address | 6,850x | 26.1x |
| Quoted text | 13,400x | 6.5x |
| JSON object | 7,240x | 33.6x |

Table 2: Speed measurements, expressed as relative speedups of this paper over Outlines (Willard & Louf, 2023). The constraints are detailed in Appendix C.4.

expressions for this form of depth-truncated JSON. Naturally, one can also directly convert the JSON grammar into a PDA-based constraint.

### 5.2 Python dataclasses

Another case study is constraining LMs to output Python dataclasses (Van Rossum & Drake, 2009). Like the JSON schemas described above, a Python dataclass defines a mapping from field names to types, and we have implemented similar tooling that reflects on a dataclass to automatically build a regular expression matching constructor calls. This enables a particularly convenient API where one can use a dataclass type `T` to build a constraint, and then parse the constrained LM response back into an instance of `T`.

### 5.3 Speculative decoding

Speculative decoding (Leviathan et al., 2023) is a method for speeding up LM inference via speculative execution. Short runs of tokens are sampled from a faster *approximation model*, and then verified with a more accurate *target model* that accepts some prefix of the sampled tokens. Although the system now runs two LMs instead of one, significant latency reductions can still be achieved due to batching and parallelization of the target model. The speedup is largely determined by the *acceptance rate*, which is the proportion of speculatively-sampled tokens that pass verification.

The approximation model is typically a smaller LM that has disproportionately greater difficulties following formal syntax, so a natural application is to constrain the approximation model and increase its acceptance rate on formal language outputs. There are several complications to address. For one, the constraint must be "rewindable" because the target model can reject some of the speculated tokens, forcing the approximation model to restart at an earlier point. This can be accomplished by tracking a full history of states instead of just one. In addition, the logits of the target model must also be penalized, or the divergence between the two will reduce the acceptance rate and may even allow the target model to resample invalid tokens. To avoid re-evaluating the constraints, we devised a method for sharing the penalties with the target model.

## 6 Experiments

Here we provide an empirical evaluation of the speed and correctness of our approach.

### 6.1 Speed

We measured our system and Outlines (Willard & Louf, 2023) on the following tasks:

1. **Constraint compilation**: Time to convert a regular expression into an automaton that can consume decoder tokens. See Appendix C.2 for details.
2. **Per-step overhead**: Time to apply constraints on each decoding step. See Appendix C.3 for details.

We used Gemma (Team et al., 2024), executed both systems on the same workstation, and averaged over 10 runs to reduce noise. Both systems were timed on several regexes reflecting different usage regimes and applications. Since runtime can vary based on platform, instead of giving precise latencies we report scaling factors relative to Outlines; see Table 2.

| Model | Unconstrained | Constrained |
|-------|---------------|-------------|
| Gemma-2B | 0.09 | 1.0 |
| Gemma-7B | 0.65 | 1.0 |

Table 3: Proportion of responses that conform to the required output schema, for Gemma models of different sizes, with and without constraints.

To help ground the comparison, here are some rough latency ranges. For constraint compilation, Outlines takes 10s of seconds on JSON and a few seconds elsewhere, while our system takes a few milliseconds on JSON and 100s of microseconds elsewhere. For per-step overhead, Outlines takes 10s of microseconds and our system takes a few microseconds.

While our system has less per-step overhead, in practice both systems have negligible latency so the speedup is not very impactful. The compilation speedup, on the other hand, is a qualitative difference that lowers the barriers to entry for applying constraints and enables new usage patterns—for example, pre-compilation is no longer required.

The primary reason for the large compilation speedup is that we use OpenFST's highly-optimized[4] implementations of FST operations (Riley et al., 2009). Critically, this is only possible because we reformulated the whole problem in terms of automata. As an analogy, imagine two programs for some scientific computing task. The first is entirely written by hand, while the second reduces the problem to linear algebra and uses NumPy (Harris et al., 2020). The latter is likely to be much faster, due to the sheer volume of effort that has gone into optimizing NumPy.

### 6.2 Correctness

We exercise output formatting correctness with Gemma on GPQA (Rein et al., 2023); see Table 3 for results and Appendix D for additional details. The Gemma models have issues following the schema without constraints, but achieve perfect conformance with constraints.

## 7 Conclusion and future work

In this paper, we described a system for constraining LM decoding. Our key contribution is a reformulation of detokenization as an FST, which enables our other contributions by bringing the entire task of constrained decoding into the domain of automata. Although the problems raised by ambiguous and misaligned tokenizations are quite thorny, we derive simple, elegant, and highly-performant solutions by leveraging the considerable toolkit of automata theory.

In future work, we would like to explore PDAs more deeply, as thus far we have focused on FSAs because they appeared to be the more robust and scalable option. One particular issue is that the grammar must be written carefully to avoid yielding a non-deterministic PDA. This is a well-studied problem, and there are similar limitations regarding grammar specification in parsers for determinstic context-free languages: as a simple example, an $LL(k)$ parser cannot parse a left-recursive grammar. There is a deep literature characterizing deterministic grammars and PDAs that seems like it would have many useful insights for avoiding unnecessary non-determinism (Greibach, 1965; Koch & Blum, 1997; Harrison & Havel, 1973; Havel & Harrison, 1974; Geller et al., 1976, among others).

Other avenues of exploration for PDAs include use cases and efficiency of representation. When decoding from an LM, one generally specifies a finite maximum budget of decoded tokens, and that limit shrinks the gap between context-free and regular languages. For example, if we decode at most 100 characters then the language $a^n b^n$ is no longer context-free, since we can bound $n \leq 50$. It may be interesting to investigate longer-running, possibly multi-phase constraint use cases where PDAs can provide more benefit. On the other hand, even for shorter outputs where FSAs are competitive, PDAs might still provide value because they are generally more compact than equivalent FSAs.

---

[4]Even further optimization is possible, too. For example, Argueta & Chiang (2018) claim a 4.5x improvement over OpenFST by applying GPUs.

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

## A   Tokenization mismatch problems

Suppose that there are two APIs that can be called like so: `foo(123)` or `bar(456)`, and that each call has two possible tokenizations, with the following LM scores:

| Score | Tokenization |
|---|---|
| 0.4 | `fo｜o(1｜2｜3)` |
| 0.3 | `bar｜(｜456｜)` |
| 0.2 | `foo｜(｜123｜)` |
| 0.1 | `ba｜r(4｜5｜6)` |

When all tokenizations are allowed, `foo(123)` is clearly the best LM output. However, tokens like `r(4` are difficult to support because they cross boundaries between different syntactic units. Handling all possible cases in a manual implementation is difficult to the point of being impractical. It's worth emphasizing that this is a real problem in current LMs. For example, the Gemma (Team et al., 2024) vocabulary includes tokens like `.")`, or `==""){`, which span across several syntactic units.

A naïve constraint system might simply force the LM tokenization to fit the formal syntax, effectively excluding the first and last rows in the table above. This simplifies the implementation, but in this example it also inverts the ranking—of the remaining rows, `bar(456)` is the best LM output. Thus, forcing the tokenization to fit the formal language can distort scoring in harmful ways.

### A.1   Grammar-based constraints in Outlines are over-constrained

As of this writing, grammar-based constraints in Outlines (Willard & Louf, 2023) do not allow LM tokens to cross between terminals in the grammar. This can lead to scoring distortions as described above, as their system will forbid certain tokenizations.

Reprising the example from earlier, consider the following simple CFG:

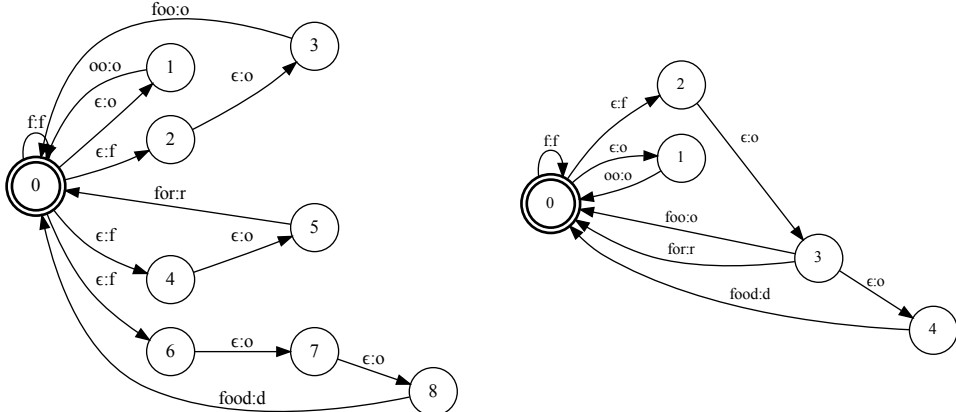

Figure 9: The FST constructed by Algorithm 1 from the vocabulary in Figure 4 (left), and the equivalent compact trie-like FST copied from Figure 4 for convenient reference (right). Note, for example, that there are three "$\epsilon$:f" edges on the left, which are merged into a single edge on the right.

$$
\begin{array}{rcl}
\text{S} & \rightarrow & \text{FUNCTION "(" NUMBER ")"} \\
\text{FUNCTION} & \rightarrow & \text{"foo" | "bar"} \\
\text{NUMBER} & \rightarrow & \text{"123" | "456"}
\end{array}
$$

This grammar follows typical practice and defines separate terminals for brackets like (, identifiers like foo, and literals like 123. If LM tokens are not allowed to cross between terminals, then tokens like r(4 will be forbidden. This can lead to a ranking inversion as described above, where the best LM output changes depending on whether tokenization is free or restricted.

# B  Proofs of correctness

Here we prove the correctness of our approach.

## B.1  Proof of Algorithm 1

In this section we prove that, given a vocabulary $V$, Algorithm 1 constructs an FST that transduces any token sequence into its detokenization. Recall that there are two versions of the detokenizing FST: the "original" structure as constructed by Algorithm 1, and the "compact" structure where equivalent initial edges of each cycle have been merged. See Figure 9 for examples of both.

We first prove correctness for the original detokenizing FST, and then briefly extend to the compact detokenizing FST.

### B.1.1  Correctness of the original detokenizing FST

Formally, let $V$ be a set of opaque tokens, where each token $v \in V$ can be detokenized into a unique character sequence $d(v) \in \Sigma^*$. Let $W = \{d(v) : v \in V\}$ be the set of detokenizations of each token. The detokenization of a token sequence $x \in V^*$ is the concatenation of the detokenizations of each token: $D(x) = d(x_1)d(x_2)\cdots d(x_n)$, where $n = |x|$. Let $\mathcal{T}_V$ be the FST as originally constructed from $V$ by Algorithm 1. Recall that the notation $\mathcal{T}_V(x)$ denotes the output sequence generated by that FST for the input $x$—our goal is to prove $\mathcal{T}_V(\cdot) \equiv D(\cdot)$.

We now review the structure of $\mathcal{T}_V$; see Figure 9 (left) for a visual example. First, note that $\mathcal{T}_V$ has exactly one initial and final state, which are the same "root" state $q_r$. For each $v \in V$, Algorithm 1 adds exactly one cycle that starts and ends at $q_r$ and generates the characters of $d(v)$ as outputs, in order. The last edge of each cycle consumes input $v$, while the other edges consume nothing (i.e., input $\epsilon$). There are no edges outside of those $|V|$ cycles.

Before proving the main theorem, we first prove some supporting results. First, we define the possible paths one may traverse through $\mathcal{T}_V$.

**Lemma 1.** *Every valid transduction by $\mathcal{T}_V$ starts and ends at $q_r$, and traverses zero or more of the $|V|$ cycles, in any order.*

*Proof.* By definition of $I_V$ and $F_V$, every transduction must start and end at $q_r$. A trivial transduction stops immediately at $q_r$, traversing no edges or cycles. Any traversal that leaves $q_r$ must enter one of the $|V|$ cycles and cannot stop until it returns to $q_r$ by completing the cycle—an arbitrary number of cycles may be traversed in this manner. Finally, since all cycles are start and end at $q_r$, there are no ordering dependencies: any cycle may be traversed after any other cycle. Therefore, a transduction by $\mathcal{T}_V$ must traverse any number of cycles in any order. ∎

From this we derive several related corollaries, which we prove *en masse*.

**Corollary 1.** *The domain of $\mathcal{T}_V(\cdot)$ is $V^*$.*
**Corollary 2.** *The codomain of $\mathcal{T}_V(\cdot)$ is $W^*$.*
**Corollary 3.** *For any $x \in V^*$, $\mathcal{T}_V(x)$ is $D(x)$.*

*Proof.* Each result is immediate from Lemma 1 and the fact that each of the $|V|$ cycles in $\mathcal{T}_V$ uniquely consumes exactly one input token $v$ and outputs exactly the characters of $d(v)$. ∎

These combine to prove the main theorem.

**Theorem 1.** *For any vocabulary $V$, $\mathcal{T}_V(\cdot) \equiv D(\cdot)$.*

*Proof.* First, observe that $D(\cdot) : V^* \to W^*$ has the same domain and codomain as $\mathcal{T}_V(\cdot)$. Two functions are equivalent if they have the same domain (Corollary 1), codomain (Corollary 2), and map the same inputs to the same outputs (Corollary 3). ∎

### B.1.2 Correctness of the compact detokenizing FST

We briefly extend the proof above to the compact detokenizing FST shown in Figure 4 and Figure 9 (right). This FST is constructed from the original detokenizing FST by recursively merging equivalent initial edges of each cycle, resulting in a structure similar to a prefix trie.

First, note that the last edge in each of the $|V|$ original cycles is never merged, because each final edge consumes a unique input token. The compact detokenizing FST can thus be partitioned into two halves: a prefix trie over the first $|v| - 1$ characters of every token, and $|V|$ edges cycling back from the leaves of the trie to $q_r$.

It follows that there are still exactly $|V|$ cycles in the compact FST, where each cycle consists of a path from $q_r$ to a leaf of the prefix trie followed by one of the $|V|$ edges back to $q_r$. Stated another way, all of the $|V|$ original cycles still exist, but instead of being disjoint they now share some edges. Therefore, Lemma 1 still holds and the rest of the proof follows.

### B.2 Proof of Algorithm 2

This proof proceeds in two phases: we first show that $\mathcal{A}_{R \circ V}$ accepts exactly the token sequences we want, and then briefly argue that the constraint application in the decoding loop is correct.

### B.2.1 Correctness of $\mathcal{A}_{R \circ V}$

For brevity, we reuse the notation set up in Appendix B.1.1 instead of restating it. Recall that we overload notation so $\mathcal{A}_{R \circ V}(x)$ is a predicate indicating whether $\mathcal{A}_{R \circ V}$ accepts $x$. The proof is a straightforward application of existing results.

**Theorem 2.** *For any $x \in V^*$, $\mathcal{A}_{R \circ V}(x)$ is true iff $D(x)$ matches the regex $R$.*

*Proof.* First, note that $\mathcal{A}_{R \circ V}$ is the composition of $\mathcal{A}_R$ and $\mathcal{T}_V$. From previous work establishing the connection between regexes and FSAs (Kleene, 1951; McNaughton & Yamada, 1960; Thompson, 1968), we know that for any character sequence $w \in \Sigma^*$, $\mathcal{A}_R(w)$ is true iff $w$ matches the regex $R$. From the definition of FST-FSA composition (Riley et al., 2009), we

know that for any $x \in V^*$, $\mathcal{A}_{R \circ V}(x) = \mathcal{A}_R(\mathcal{T}_V(x))$. Finally, by applying Theorem 1 we have that $\mathcal{A}_{R \circ V}(x) = \mathcal{A}_R(D(x))$. ∎

Stated plainly, $\mathcal{A}_{R \circ V}$ accepts all token sequences that, when detokenized, match the regex $R$.

### B.2.2  Correctness of constraint application

Prior work by Deutsch et al. (2019) and Zhang et al. (2019) already developed a framework for applying token-accepting FSA constraints (called "covering grammars" in the latter) to sequence models, so we merely sketch a proof here.

The decoding loop is augmented with a state $q$, initially $I_{R \circ V}$, that tracks the current state in the constraint FSA (a single state suffices because $\mathcal{A}_{R \circ V}$ is determinized). On each decoding step, we mask out the sampling logits of any token not among the outbound edges of $q$, and then update $q$ based on the decoded token. $\mathcal{A}_{R \circ V}$ accepts the decoded LM output by construction, and from Theorem 2 we know that the detokenized LM output matches the regex $R$. In addition, since Algorithm 2 does not touch the logits of non-masked tokens, the LM is free to output any token sequence that matches $R$.

### B.3  Proof of Algorithm 3

As in the previous section, we first prove correctness of $\mathcal{P}_{G \circ V}$ and then briefly argue for the correctness of the constraint application. As Algorithm 3 is a minor variation on Algorithm 2, the proofs and arguments are very similar.

### B.3.1  Correctness of $\mathcal{P}_{G \circ V}$

For brevity, we reuse the notation set up in Algorithm B.1.1 instead of restating it. Recall that we overload notation so $\mathcal{P}_{G \circ V}(x)$ is a predicate indicating whether $\mathcal{P}_{G \circ V}$ accepts $x$. As with Theorem 2, the proof is a straightforward application of existing results.

**Theorem 3.** *For any $x \in V^*$, $\mathcal{P}_{G \circ V}(x)$ is true iff $D(x)$ matches the grammar $G$.*

*Proof.* First, note that $\mathcal{P}_{G \circ V}$ is the composition of $\mathcal{P}_G$ and $\mathcal{T}_V$. From previous work (Allauzen & Riley, 2012, Section 4.5), we know that for any character sequence $w \in \Sigma^*$, $\mathcal{P}_G(w)$ is true iff $w$ matches the grammar $G$. From the definition of FST-PDA composition (Allauzen & Riley, 2012, Section 4.2), we know that for any $x \in V^*$, $\mathcal{P}_{G \circ V}(x) = \mathcal{P}_G(\mathcal{T}_V(x))$. Finally, by applying Theorem 1 we have that $\mathcal{P}_{G \circ V}(x) = \mathcal{P}_G(D(x))$. ∎

### B.3.2  Correctness of constraint application

The decoding loop in Algorithm 3 is essentially the same as Algorithm 2, but with minor changes to traverse a PDA instead of an FSA. Specifically, whereas Algorithm 2 only tracks an FSA state $q$, Algorithm 3 tracks a PDA state $q$ and a stack $s$, initially empty (a single $q$ and $s$ suffice, because we assume $G$ is a deterministic context-free grammar). When considering the outbound edges of $q$, each edge $e$ is also filtered based on whether the current stack $s$ matches $e^\uparrow$, the stack symbols popped by $e$. Finally, when an edge is traversed, the stack $s$ is also updated by popping $e^\uparrow$ and pushing $e^\downarrow$.

By traversing $\mathcal{P}_{G \circ V}$ during decoding and masking logits according to the outbound edges allowed by $q$ and $s$, the LM is forced to generate a token sequence that $\mathcal{P}_{G \circ V}$ accepts. Therefore, by Theorem 3 the detokenized LM output must match the grammar $G$. At the same time, since Algorithm 3 does not touch the logits of non-masked tokens, the LM is free to generate any output that matches $G$. In particular, the LM is allowed to output tokens that span multiple terminals or non-terminals (see Figure 7).

## C  Speed measurement details

This section provides additional details about the experiments we ran to measure speed.

### C.1  Outlines setup

Following the instructions on the Outlines (Willard & Louf, 2023) homepage, we downloaded and installed Outlines using the command `pip install outlines`. The specific version we received was Outlines v0.0.34.

We accessed the Gemma (Team et al., 2024) model via its integration into Outlines, using the command `outlines.models.transformers("core-outline/gemma-2b-instruct")`. Note that for these experiments, the LM's parameter size does not matter because we are timing computations outside the LM. Only the vocabulary size matters, and the vocabulary is constant across Gemma sizes.

## C.2    Constraint compilation

For our system, compilation includes the following operations:

1. Parsing the regex into an FSA.
2. Optimizing the regex FSA (e.g., determinization).
3. Composing the FSA with the vocabulary FST.
4. Optimizing the composed FSA.

For Outlines, we were not familiar with that codebase and did not want to inadvertently include or exclude the wrong operations. Instead of trying to isolate the exact function(s) to benchmark, we took a black-box approach. We first called `outlines.generate.regex()` on the trivial regex `/x/` to establish a baseline latency. Then we called it again on the target regex and subtracted off the baseline latency. In this way, we hoped to exclude any fixed costs that do not scale with regex complexity.

This baseline correction is significant, as the trivial regex latency was 1/4 to 1/3 as large as most target regex latencies. For consistency, we performed the same baseline subtraction on our system, though it had far less impact—around 1/20 of most target regex latencies.

For both systems, we processed a warm-up regex before starting the timed runs in order to trigger any lazy initialization. This is important in Outlines, because the vocabulary index is lazily initialized and takes tens of seconds to build.

### C.2.1    Caching in Outlines

Recall that for each regex, we timed 10 runs and averaged the results to reduce noise. Outlines has a regex-to-indexed-FSM cache, however, so naïvely the last 9 runs would be cache hits with inaccurate latency. We addressed this by calling `outlines.disable_cache()` to force the full computation on every run. Based on our (non-expert) inspection of the codebase, the only relevant cache this disables is the regex-to-indexed-FSM cache.

As a sanity check, we also tried leaving caching on and instead appended a unique dummy character to the target regex on each run (e.g., `/foo0/`, `/foo1/`, etc.). This effectively "disables" the regex-to-indexed-FSM cache without impacting any other non-obvious caching in the system. These timings were very similar and slightly slower, likely due to the extra dummy character. Thus we infer that `outlines.disable_cache()` does not cause a significant distortion of our measurements. The reported figures in Table 2 are based on measurements with the cache disabled, rather than the dummy character approach, as they were faster and did not require us to mangle the regex.

## C.3    Per-step overhead

For both systems, per-step overhead includes the following operations:

1. From the start state of the automaton, find the set of valid next tokens.
2. Use these to build a vector marking valid and invalid tokens.
3. Advance the automaton to the state corresponding to the first valid next token.

We initially tried a more end-to-end approach, where we ran a Gemma LM with and without constraints and subtracted to measure the overhead. Unfortunately, LM latency is several orders of magnitude larger than the constraint overhead, so the measurements are dominated by noise in the LM computation—often yielding negative overhead measurements. Therefore, we measured the operations above in isolation from the LM.

## C.4    Constraint regexes

Here, we describe the constraints used in the speed experiments.

### C.4.1 Multiple choice

This matches a set of predefined texts:

```
/Red|Orange|Yellow|Green|Blue|Indigo|Violet/
```

This example represents the simplest type of constraint that one might realistically want to apply. For example, this could be used for basic `/Yes|No/` question answering, or classification tasks like `/Sports|Politics|Weather/`.

### C.4.2 ISO date-time

This matches an ISO 8601-style date time expression:

```
/\d{4}-[01]\d-[0-3]\dT[0-2]\d:[0-5]\d:[0-5]\d([+-][0-2]\d:[0-5]\d|Z)/
```

This example represents a slightly more complex constraint, with a non-trivial but relatively straightforward lattice structure.

### C.4.3 IP address

This matches an IPv4 address and is copied from the Outlines documentation[5]:

```
/((25[0-5]|2[0-4]\d|[01]?\d\d?)\.){3}(25[0-5]|2[0-4]\d|[01]?\d\d?)/
```

This example is moderately complex and exercises grouped repetition. Beyond those characteristics, we primarily chose this constraint because, as it comes from Outlines, it is clearly not a "cherry-picked" example.

### C.4.4 Quoted text

This matches a double-quoted string like `"foo"`. For our system, we use the following extension-based regex:

```
/(?P<QUOTED_TEXT>)/
```

and for Outlines (Willard & Louf, 2023), we use the following equivalent regex:

```
/" *(?:[^\s"\\]|\\["n\\])(?: |[^\s"\\]|\\["n\\])*"/
```

This example is intended to highlight the extensions we presented in Section 2.5. The comparison in Table 2 shows a clear advantage relative to the other constraints.

### C.4.5 JSON object

This matches a JSON object that conforms to a JSON schema (see Section 5.1). Both our system and Outlines have methods for converting a JSON schema into a regex. The resulting regexes are quite verbose, however, so we specify the schema instead of the full regex:

```
{
  "type": "object",
  "properties": {
    "name": {"type": "string"},
    "class": {
      "type": "string",
      "enum": ["Warrior", "Rogue", "Sorceror"]
    },
    "life": {"type": "integer"},
    "mana": {"type": "integer"},
    "equipment": {
      "type": "array",
      "items": {
```

---

[5] https://outlines-dev.github.io/outlines/reference/regex/

```
      "type": "object",
      "properties": {
        "name": {"type": "string"},
        "durability": {"type": "integer"},
        "quality": {
          "type": "string",
          "enum": ["Normal", "Magic", "Unique"]
        }
      }
    }
  }
 }
}
```

This example represents a fairly complex constraint, as it involves enums, arrays, and nested sub-objects. However, we commonly apply our system to much more complex constraints.

## D   Correctness measurement details

We used Gemma (Team et al., 2024) zero-shot with the following prompt:

```
You will be given a multiple choice question with different options such as
A, B, C, D. Think step by step before giving a final answer to this question.
Format your answer with the following JSON format which uses double quotes:
```json
{
  "Final Answer": "X",
  "Solution": "step-by-step reasoning"
}
```

where X is the correct answer choice. If none of the options match, choose
the closest option as the final answer.
```

