# OpenReview forum: "Automata-based constraints for language model decoding"
_colmweb.org/COLM/2024/Conference — COLM_

### Official Review · Reviewer_mjW2 · 2024-04-23

**Rating:** 6
**Confidence:** 4
**Ethics Flag:** 1

**Summary:**

Paper authors propose an approach to force the LLM output to adhere to a given regular expression or a deterministic grammar. According to the paper, one important problem when this task should be carried out is how to handle the misalignment between the requested constraints and text tokenization. In order to solve this problem, the detokenization task is formalized as a finite state automaton. Such automaton is then composed with another automaton reflecting the regular expression or deterministic grammar that will be used to constrain the output of the LLM. The paper also enumerates some possible applications of the proposed LLM output guidance system.

I think the paper addresses a very important and timely problem within the subfield of LLMs. I also find the proposed approach elegant and effective in solving the LLM's output guidance task. However, I think there are some problems with the paper that I would like to summarize below:

- The paper is strongly focused on describing the theoretical aspects of the proposed techniques but there are no examples that help the reader to really understand the motivation of the proposal. For instance, I miss a few or at least one example of LLM guidance where tokenization can result in a failure of the system and how the proposed technique is able to successfully remove those problems.

- The paper lacks an experimentation section measuring the performance of the proposal. Also such performance is not compared with that of previously existing approaches. Authors do mention that there is a strong relationship between their work and the work carried out by Willard & Louf, 2023, but no comparison is carried out. The authors state that their automata-theoretic-based technique makes "much easier to apply finite-state automata and generalize to push-down automata", but I think the paper does not clarify which are exactly the advantages that are introduced. Even if a systematic comparison is not reported, I think that at least some comparison examples would be needed to understand the relative merits of the proposal. Something similar to this is done in [Willard & Louf, 2023] and I think is very useful to have a better idea of the proposed work.

- I miss that the paper provides algorithms describing the two main LLM guidance techniques that are proposed. Instead, those techniques are described as enumerations in the paper's text. In spite of the fact that those enumerations are understandable, I think that using algorithms would be more suited in this case to explain the techniques in a precise and unambiguous way.

I am ambivalent about this paper. On one hand, I really like the elegance of the proposal, but on the other hand, the lack of a meaningful comparison with other works beyond what is stated in the Related Work section makes difficult for the reader to understand how the proposed technique improves previously existing ones.

**Reasons To Accept:**

- The problem of LLM's output guidance is elegantly casted as a search process constrained by the composition of pairs of automata.

- The proposed technique specifically addresses the problems introduced by tokenization steps when guiding the LLM output, which has not previously been done in the literature.

**Reasons To Reject:**

- The paper mentions related work but there is no experimental section nor any comparison between the proposed technique and previously existing ones, in particular the technique proposed in [Willard & Louf, 2023].

- The paper does not include examples that would facilitate a better understanding of the problems being addressed.

---

> ### Author Rebuttal · Authors · 2024-05-29
>
> Thank you for the review!
>
> Regarding examples, there is a motivating example in the introduction that discusses the problems with tokenization.  We will sharpen or expand the example to make the issues clearer.
>
> Suppose that we have two APIs that can be called like so: `foo(123)` or `bar(456)`.  Now suppose each call has two possible tokenizations, with the following LM scores:
>
> | Score | Tokens |
> |---|---|
> | 0.4 | `fo o(1 2 3)` |
> | 0.3 | `bar ( 456 )` |
> | 0.2 | `foo ( 123 )` |
> | 0.1 | `ba r(4 5 6)` |
>
> Tokens like `r(4` are painful to support in constraints, so we might naively **force** the tokenization to fit the formal syntax.  This simplifies the implementation but excludes the first and last rows.  Of the remaining rows, `bar(456)` is the best, but with full tokenization freedom `foo(123)` would win.  Thus, forcing tokenization **distorts** scoring.
>
> We started with the naive approach long ago, but scoring distortions easily arise with data-driven LM tokenizers.  If you **don't** force the tokenization, however, you must deal with the mismatch between token boundaries and formal syntax—we solve this with automata.
>
> We ran latency and quality evals.  Full results will be added to the paper, summary below.
>
> Latency: We ran on Gemma tokenization, on the same workstation, averaging 10 runs.  Constraint compilation (regex => composed FSA) takes **100s of microseconds**, and overhead per decode step is **single-digit microseconds**.  Versus Outlines, we compile **7,000x-14,000x** faster and have **6x-25x** less step overhead.  These speedups enable applications that Outlines cannot support.
>
> Quality: We test the capability of output formatting with Gemma using [GPQA](https://arxiv.org/abs/2311.12022) by adding `"""Format your answer with the following JSON format which uses double quotes: ```json\n{ "Final Answer": "X", "Solution": "step-by-step reasoning" }\n```"""` to the preamble where X is one of the choices. Gemma2b and Gemma7b give **0.09** and **0.65** accuracy on schema-following without guided decoding and **1.0** with our constraints.
>
> One qualitative difference from Outlines is that our approach is **modular**: any tokenizer expressible as an FST can be **composed** with any constraint expressible as an automaton.  The "indexing" algorithm used in Outlines is non-trivial to extend to either PDAs (with full tokenization freedom) or new tokenizing automata as in arxiv.org/abs/2405.07671.
>
> Finally, we will rewrite the algorithms in pseudo-code.

---

> > ### Comment · Reviewer_mjW2 · 2024-05-31
> > **Rebuttal Reponse**
> >
> > Dear authors,
> >
> > thanks for your careful response. I was aware of the example provided in the introduction. However, what I was missing in the paper is a specific example of how your technique succeeds in constraining the LLM output where another previously existing one fails. This could be shown for instance for the Willard & Louf, 2023 technique.
> >
> > After reading your rebuttal, I was wondering if the scoring distortion problems you mention would apply to the Willard & Louf's technique.

---

> > > ### Author Response · Authors · 2024-05-31
> > > **Refining the example**
> > >
> > > Sorry for our misunderstanding.  Our approach and Outlines have the **same guarantees for regular expressions**, so we cannot provide an example where either system is better than the other.  Our approach is **strictly better for grammars**, because Outlines [prevents tokens from crossing (non)terminal boundaries](https://github.com/outlines-dev/outlines/issues/684), whereas we do not.
> > >
> > > As an example of where this might cause issues for Outlines but not our system, reconsider the example above, but now in the context of a CFG like:
> > >
> > > ```
> > > S => FUNCTION "(" NUMBER ")"
> > > FUNCTION => "foo" | "bar"
> > > NUMBER => "123" | "456"
> > > ```
> > >
> > > Since `foo`, `(`, `123`, and `)` are separate terminals, LM scoring with Outlines **would be distorted** in the manner described above, where the first and last rows are excluded and the LM scoring after constraints inverts the ranking of `foo(123)` vs `bar(456)`.  In contrast, our approach does not have any restrictions on (non)terminal boundaries, so LM scoring is **not distorted**.
> > >
> > > Note that the grammar above is not contrived or specially-constructed to cause problems—in programming language grammars it is natural to treat parentheses, identifiers, and numbers as separate terminals.
> > >
> > > One last note is that even if we had full parity with Outlines in terms of guarantees or correctness, our system is still more **generic/modular** and **much faster**.

---

> ### Comment · Reviewer_mjW2 · 2024-06-06
> **New Response**
>
> I deeply appreciate the careful responses of the authors. I think they have made a nice work with the paper as I stated in my initial review. However, when analyzing the messages exchanged in the rebuttal period, I continue thinking that the paper in its current state does not include important empirical results demonstrating the improvements introduced by their technique, and in my view, their inclusion would require to conduct a new review process. Because of this, I cannot modify the score I have assigned to the paper. In any case, I strongly encourage the authors to introduce the new data into the paper, since this will result in a really nice work.

---

### Official Review · Reviewer_iNwk · 2024-04-24

**Rating:** 5
**Confidence:** 3
**Ethics Flag:** 1

**Summary:**

This paper explores constraining the output of an LLM using finite state techniques, since many constrained languages can be defined in terms of formal languages (either a context-free language or even a regular language).  It also notes that tokenization issues can be accounted for by simply creating a finite state transducer to shift from the tokenized language to the output language in which the constraints are defined, and this transducer and finite state machine (or in fact, a context free equivalent called a pushdown automata) can be composed and evaluated efficiently.  The authors also suggest some "wildcard" extensions to cover common use cases, such as a paragraph of text or any single token.

The authors make a reasonable case for their approach, and is generally quite clear and accessible for a mathematically adept audience.  The case they make for using an FST to convert between languages is a useful observation.  One slight complaint is that the description of PDA is a little dense --  the example could be explained a little more -- and it is hard to understand if/how the switch to context free languages affects the efficiency, etc.

The biggest issue with the paper is the omission of an "experiments" section, where the method would be empirically evaluated on real data.  The suggested approach could be measured in terms of memory use, time/compute resources, etc. compared to other proposals or even a simple baseline (checking and rejecting non-accepted output, needing to ask the model again).  Certainly it could be compared with some of the related work, such as Outlines which has a github page with an available implementation to test against.  The paper would also be much stronger if an example implementation was provided along with the conceptual description.

Regarding related work, it is a bit unclear to me how this project differs from Outlines.  I can see they both use Finite state automata.  This paper's authors claim that their approach is more generalizable, and link to a github issue, but it is not clear to me the substance of this difference.  Is the difference in using the FST to map between the LLM tokens and external text, as hinted to in the three contributions (listed on the first page of the paper)?

**Reasons To Accept:**

- new conceptual insights into efficiently constraining LLM output

**Reasons To Reject:**

- lack of empirical validation

- lack of clear comparison with other existing techniques (either empirical --especially when those techniques have existing implementations, or even theoricial, i.e., comparison of runtimes in terms of tokens, vocabulary size, etc.)

---

> ### Author Rebuttal · Authors · 2024-05-29
>
> Thank you for the review!
>
> We have since run both latency and quality experiments.
>
> Latency: We ran on Gemma tokenization, on the same workstation, averaging 10 runs.  Constraint compilation (regex => composed FSA) takes **100s of microseconds**, and overhead per decode step is **single-digit microseconds**.  Versus Outlines, we compile **7,000x-14,000x** faster and have **6x-25x** less step overhead.  These speedups enable applications that Outlines cannot support.
>
> Quality: We test the capability of output formatting with Gemma using [GPQA](https://arxiv.org/abs/2311.12022) by adding `"""Format your answer with the following JSON format which uses double quotes: ```json\n{ "Final Answer": "X", "Solution": "step-by-step reasoning" }\n```"""` to the preamble where X is one of the choices. Gemma2b and Gemma7b give **0.09** and **0.65** accuracy on schema-following without guided decoding and **1.0** with our constraints.
>
> We will expand the related work section w.r.t. Outlines.
>
> As the reviewer notes, the main difference is the detokenizing FST.  However, the FST itself is **not** the main value-add—the value comes from **expressing** detokenization as an FST.  This brings the **whole constraint problem** into the domain of automata, so we can apply **general algorithms** like composition instead of **bespoke algorithms** like "indexing" in Outlines.
>
> As a result, our approach is more extensible.  For example, for CFGs Outlines' indexing algorithm **forbids** tokens from crossing (non)terminal boundaries (see footnote 3).  For us, FST-PDA composition is both **straightforward** and **more functional**, as it allows crossing.
>
> As another example, arxiv.org/abs/2405.07671 proposed a DFA for "correct" BPE tokenization.  Handling BPE rules in Outlines likely **requires a rewrite** of their indexing algorithm.  For us, we simply convert that DFA into a detokenizing FST, compose it with the regex FSA, and we're done.
>
> If you fully commit to automata as we have, the problem space **decomposes** nicely.  Any tokenizer expressible as an FST can be **swapped in**.  Any constraint expressible as an automaton can be **composed on top**.  Moreover, automata have been studied since the 1950s and have a **wealth of well-optimized algorithms**—note the stark latency comparison vs Outlines.
>
> Another diff with Outlines is that we add extensions to optimize common expressions.
>
> Finally, we cannot provide full implementations, but will rewrite the algorithms in pseudo-code.

---

> > ### Comment · Reviewer_iNwk · 2024-06-05
> > **changes sound good**
> >
> > In this response, the authors do confirm what I already inferred, that their finite state composition approach was more generalizable than other approaches like that of Outlines, which is helpful.  As far as the reasons this is more useful, I do think this is covered well in the paper and I am also aware of the "wealth of well-optimized algorithms" available with automata.  Though useful to know, this clarification alone doesn't change my score of the paper.
> >
> > As far as the weakness regarding experiments, if they author do make the changes promised, this would add substantially to the paper, at least shifting it to "6: Marginally above acceptance threshold" if not higher.  However, I am not able to review such changes myself, I can only score the paper as written, so I will raise my score to "5".

---

> > > ### Author Response · Authors · 2024-06-06
> > > **not allowed to submit revision during discussion**
> > >
> > > We thank the reviewer for taking the time to go through our rebuttal and increasing the score. We would love to update the submission to reflect our changes but according to the email sent by PCs titled **[COLM] Clarification regarding rebuttals + camera ready page limit**
> > > ```
> > > We do not allow to revise the submitted content during the rebuttal and discussion period, and we limit the length of rebuttals.
> > > ```
> > > We will reflect our changes in the final submission as promised. We hope this is understandable. Thank you!

---

> ### Author Response · Authors · 2024-06-03
> **Kind reminder on the close date of discussion is approaching**
>
> Dear reviewer iNwk,
> Thank you for the constructive suggestions! We would like to kindly raise your awareness that the close date of discussion is approaching. We hope our rebuttal have addressed most of the concerns and if not happy to provide more clarification.

---

### Official Review · Reviewer_CmqW · 2024-04-26

**Rating:** 8
**Confidence:** 4
**Ethics Flag:** 1

**Summary:**

This draft deals with the problem of constraining a LM to generate output satisfying a certain syntax that can be expressed by means of some formal grammar, as for instance the syntax of specific API calls. This problem is complicated by tokenization, which is misaligned with the formal grammar. The proposed approach is based on the idea of detokenization and uses the framework of automata theory to derive an efficient solution for the regular language case. The idea is also extended to deterministic context-free languages.

**Questions To Authors:**

No question to the authors. What follows is a list of minor suggestions.

You write sequences in the form $e_1 \ldots e_n$, but a much more used convention in formal language theory is $e_1 \cdots e_n$. There are several instances in the draft.

pg.1,l.-7: separator > operator ?

pg.2: iff e_i^\sigma = \ep. > iff e_i^\sigma = \ep, for some i.

the following bib entry seems incomplete: 'Brandon T. Willard and Remi Louf. Efficient guided generation for large language models, 2023.'.

**Reasons To Accept:**

The problem of constraining LM output to a given syntax is gaining increasing attention at time of writing. I like the formal approach of this draft, and I think the idea is original. The presentation is also very good. People will use this idea and cite this work.

**Reasons To Reject:**

None that I can see. The basic implementation of detokenization through finite automata is very simple, but I don't think this is a reason to reject.

---

> ### Author Rebuttal · Authors · 2024-05-29
>
> Thank you for the review!
>
> Thanks for reading the paper so carefully and finding those issues, we will be sure to fix them in the paper.
>
> One note on detokenization.  We agree that the current version is very simple, but our system generalizes to any tokenizer expressible as an FST.  For example, we recently saw arxiv.org/abs/2405.07671, which presents a construction for a DFA that represents "correct" BPE tokenizations.
>
> We can easily convert their DFA into a detokenizing FST—just replace each token arc with a chain of `character_i:epsilon` arcs (`i` from `1` to `n-1`) followed by a `character_n:token` arc.  The rest of our system then works as-is, by composing a regex FSA or grammar PDA on top of that FST.
>
> We would also like to highlight some experiments we ran on latency and quality.
>
> Latency: We ran on Gemma tokenization, on the same workstation, averaging 10 runs.  Constraint compilation (regex => composed FSA) takes **100s of microseconds**, and overhead per decode step is **single-digit microseconds**.  Versus Outlines, we compile **7,000x-14,000x** faster and have **6x-25x** less step overhead.  These speedups enable applications that Outlines cannot support.
>
> Quality: We test the capability of output formatting with Gemma using [GPQA](https://arxiv.org/abs/2311.12022) by adding `"""Format your answer with the following JSON format which uses double quotes: ```json\n{ "Final Answer": "X", "Solution": "step-by-step reasoning" }\n```"""` to the preamble where X is one of the choices. Gemma2b and Gemma7b give **0.09** and **0.65** accuracy on schema-following without guided decoding and **1.0** with our constraints.

---

> ### Comment · Reviewer_CmqW · 2024-06-05
> **Theory vs. experiments**
>
> I like the theoretical contribution in this draft, and I am convinced it will stimulate further work in this direction on the part of the community.  I don't feel experimental assessment is needed in this case, therefore I still stand by my original score.

---

> > ### Author Response · Authors · 2024-06-06
> > **appreciate recognizing the main point**
> >
> > We thank the reviewer for spending the time to reply and recognizing the main point we try to convey in this draft.

---

### Official Review · Reviewer_wViX · 2024-04-29

**Rating:** 5
**Confidence:** 4
**Ethics Flag:** 1

**Summary:**

The paper presents a system for restricting LM decoding to guarantee correct statements in a particular formal language like json expressions or python coding. The proposal is to use a grammar written as regular expressions, but before it is necessary getting units of the language vocabulary out of the subwords delivered because input has been tokenized into subwords. Finite state automatas are proposed to handle this.

**Reasons To Accept:**

The presented work aims to improve the applicability of language models

**Reasons To Reject:**

The contribution of the paper in relation to the areas covered by the call is not clear, because the paper does not address the language modelling.  The use of regular expressions to handle the output of the model is not new, as the authors recognized.
Authors propose to use FST for detokenization, but it looks as is only the reconstruction of acceptable terms out of the subtokens that the model delivers as output. There is no evaluation of the requirements of building the FST for a particular language. There is no consistent evaluation of the proposal made in this paper that could help to see the contribution.  In fact, authors acknowledge that their contribution is not empirical, but conceptual.

---

> ### Author Rebuttal · Authors · 2024-05-29
>
> Thank you for the review!
>
> We believe our paper matches the selected areas: **inference algorithms** and **tools and code**.  Constraints are applied at inference time in the decoding loop, and can enforce formatting of tool API calls or code snippets. Our new experiment shows prompting alone can't guarantee proper format.
>
> Both we and Outlines use regexes, but our use of automata theory is deeper and more general.  They handle detokenization with a **bespoke "indexing" algorithm**, whereas we reformulate detokenization as an FST.  Although the FST is simple, it crucially brings the **whole constraint problem** into the domain of automata, so we can apply **general algorithms** like composition instead of **bespoke algorithms** like Outlines.
>
> As a result, our approach is more extensible.  For example, for CFGs Outlines' indexing algorithm **forbids** tokens from crossing (non)terminal boundaries (see footnote 3).  For us, FST-PDA composition is both **straightforward** and **more functional**, as it allows crossing.
>
> As another example, arxiv.org/abs/2405.07671 proposed a DFA for "correct" BPE tokenization.  Handling BPE rules in Outlines likely **requires a rewrite** of their indexing algorithm.  For us, we simply convert that DFA into a detokenizing FST, compose it with the regex FSA, and we're done.
>
> Another diff with Outlines is that we add extensions to optimize common expressions.
>
> The reviewer mentions "the requirements of building the FST for a particular language" but we don't follow, **please clarify**.
>
> Regarding evals, we measured latency and quality.  We will add full results to the paper, summary below.
>
> Latency: We ran on Gemma tokenization, on the same workstation, averaging 10 runs.  Constraint compilation (regex => composed FSA) takes **100s of microseconds**, and overhead per decode step is **single-digit microseconds**.  Versus Outlines, we compile **7,000x-14,000x** faster and have **6x-25x** less step overhead.  These speedups enable applications that Outlines cannot support.
>
> Quality: We test the capability of output formatting with Gemma using [GPQA](https://arxiv.org/abs/2311.12022) by adding `"""Format your answer with the following JSON format which uses double quotes: ```json\n{ "Final Answer": "X", "Solution": "step-by-step reasoning" }\n```"""` to the preamble where X is one of the choices. Gemma2b and Gemma7b give **0.09** and **0.65** accuracy on schema-following without guided decoding and **1.0** with our constraints.

---

> ### Author Response · Authors · 2024-06-03
> **Kind reminder on the close date of discussion is approaching**
>
> Dear reviewer wViX,
> Thank you for the constructive suggestions! We would like to kindly raise your awareness that the close date of discussion is approaching. We hope our rebuttal have addressed most of the concerns and if not happy to provide more clarification.

---

> > ### Comment · Reviewer_wViX · 2024-06-06
> >
> > Thanks for trying to clarify the main issues I raised in my review. I appreciate you have now empirical evaluation to assess your proposal. I changed the score, although, to me the paper is not convincing in describing the importance of the problem and the solution.

---

### Decision · Program_Chairs · 2024-07-10

**Decision:**

Accept

**Comment:**

This submission presents an interesting tokenization approach that can contribute to improving the applicability of language models. The paper makes a formal contribution, and does it well. It could be stronger by including an empirical contribution. The authors conducted some experiments during the discussion period, and the paper would be strengthened significantly by including these (and further expanding them). Beyond the empirical evidence, it will allow for better comparison to prior work.